# Scientific and Pharmaceutical Aspects of *Christensenella minuta*, a Promising Next-Generation Probiotic



Ágota Pető [1,2,†], Dóra Kósa [1,2,†], Zoltán Szilvássy [3], Pálma Fehér [1,2], Zoltán Ujhelyi [1,2], Gabriella Kovács [4], István Német [4], István Pócsi [5] and Ildikó Bácskay [1,2,*]

1. Department of Pharmaceutical Technology, Faculty of Pharmacy, University of Debrecen, Nagyerdei Körút 98, 4032 Debrecen, Hungary
2. Institute of Healthcare Industry, University of Debrecen, Nagyerdei Körút 98, 4032 Debrecen, Hungary
3. Department of Pharmacology and Pharmacotherapy, Faculty of General Medicine, University of Debrecen, Nagyerdei Körút 98, 4032 Debrecen, Hungary
4. Fluart Innovative Vaccines Ltd., Fő út 7, 2097 Pilisborosjenő, Hungary
5. Department of Molecular Biotechnology and Microbiology, Institute of Biotechnology, Faculty of Science and Technology, University of Debrecen, Egyetem tér 1, 4032 Debrecen, Hungary
* Correspondence: bacskay.ildiko@pharm.unideb.hu; Tel.: +36-52-411-717-54034
† These authors contributed equally to this work.

**Abstract:** *Christensenella minuta* (*C. minuta*), a member of a recently described bacterial family, is one of the most heritable next-generation probiotics. Many observational studies confirmed that the relative abundance of *C. minuta* is associated with lean body types with a low host body mass index (BMI), and is also influenced by age, diet, and genetics. By utilizing its benefits, it could be suited to many therapies, including human and animal health as well. However, a reliable method for culturing the strain must also be developed to enable the therapeutic administration of the microbe. Sludge microfiltration could be a promising solution for large scale-up cultivation. In this review, different processing methods are also described from pharmaceutical aspects.

**Keywords:** *C. minuta*; next-generation probiotics; anaerobic; sludge; cultivation

## 1. Introduction

The human gastrointestinal tract is colonized by a complex system of different microorganisms [1]. Beneficial gut bacteria have many important functions, such as producing numerous nutrients for the host, preventing infections caused by different intestinal pathogens, and modulating healthy immunological responses [2]. Prebiotics are described as selectively fermented ingredients, which lead to specific changes in the composition or the activity of the gastrointestinal microbiota, thereby providing benefits to the health of the host system [3]. Prebiotics belong to the class of nutrients, which are degraded by the gut microbiota. They can feed the microbiota of the intestine, and among their decomposition products, short-chain fatty acids can be found, that are released into the bloodstream; therefore, they affect not only the gastrointestinal tract but many other distant organs as well [4,5]. Probiotics are living, non-pathogenic microorganisms with beneficial physiological effects on the host system when administered in the right dose [6]. Probiotics can improve human immunity, thus preventing pathogen colonization and decreasing the incidence and severity of the occurring infections. They usually consist of *Saccharomyces boulardii* yeast or lactic acid bacteria, e.g., *Lactobacillus* and *Bifidobacterium* species, and are regulated as dietary supplements and foods. Probiotics act through numerous pathways, including lowering the intestinal pH, reducing colonization and invasion by pathogenic microorganisms, and altering the host's immune response. Probiotics are considered to be safe and well tolerated, with the side effects of bloating and flatulence [7]. They must be used cautiously in patients who are immunocompromised or those with central venous catheters

to avoid the occurrence of systemic infections [8,9]. Synbiotics are a mixture of live microorganisms and substrates utilized by the host microorganisms that result in health benefits for the host [10]. Two subsets of synbiotics can be described: complementary and synergistic synbiotics. Complementary synbiotics consist of a probiotic combined with a prebiotic, which is designed to target microorganisms, while synergistic synbiotics are specifically designed to be selectively utilized by the co-administered microorganism(s) [11,12].

The incorporation of probiotics, prebiotics, or synbiotics into the human diet has beneficial effects on the gut microbiota. In an average diet, they could be available in sufficient quantities, as they can be consumed in the form of vegetables and fruits or dairy products, but with modern eating habits, this cannot be achieved. In such cases, other sources including pharmaceuticals and functional foods are available [13,14]. However, based on previous estimates, approximately 99% of all bacteria are still uncultivable in a large scale, limiting their pharmaceutical administration [15]. *C. minuta*, a member of *Firmicutes*, is a promising next-generation probiotic struggling with the above-mentioned difficulty, even though it has a number of scientifically proven beneficial effects. The objective of this review is to summarize the different scientific aspects and pharmaceutical perspectives of *C. minuta* to increase knowledge about this microorganism.

## 2. *Christensenella minuta* as a Potential Next-Generation Probiotic

The importance of traditional probiotics has rapidly intensified over the last few decades due to their safety and easy availability in fermented foods. Concurrently, their widespread administration has made them more resistant to certain diseases [16]. Many studies confirmed that the dysbiosis of gut microorganisms is strictly associated with the development of chronic gastrointestinal inflammation, obesity, metabolic syndromes, diabetes mellitus, cardiovascular diseases, and even neurodegenerative diseases [17]. The administration of commonly used traditional probiotics such as *Bifidobacterium* spp. and *Lactobacillus* spp. does not always seem to be effective against the above-mentioned specific disorders [18]. Using these commensal bacteria as natural candidates for probiotics to maintain or restore normal homeostasis within the gastrointestinal tract revealed interest in new microbes with potential health benefits. The rapid progress in culturing procedures and innovative genome sequencing methodologies allow scientists to develop bespoke probiotics that address specific patient needs and issues [19]. Due to growing interest, a number of previously unknown microorganisms have been identified recently, referred to as next-generation probiotics (NGPs). NGPs have emerged as possible therapeutic sources for the treatment of diseases requiring specific, complementary, and combination therapies. The term usually refers to newly isolated functional species that have never been used in the food industry before and are expected to emerge as potential therapeutic sources of several diseases due to their beneficial properties [20]. Recent studies have revealed many anaerobic bacteria as potential next-generation probiotics, including different *Bifidobacterium* spp., *Prevotella copri*, *Parabacteroides goldsteinii*, *Akkermansia muciniphila*, *Bacteroides fragilis*, and *Faecalibacterium prausnitzii* [16]. Unfortunately, their extreme sensitivity to oxygen and acidic gastric conditions make them challenging to work with [21]. According to recent proof-of-concept studies, a common feature of several next-generation probiotics is short-chain fatty acid production, particularly butyrate, known to support healthy intestinal homeostasis, and improve epithelial barrier function [22].

More recently, the butyrate producer gut commensal *C. minuta* has been in the spotlight as a new candidate of next-generation probiotics [23,24]. *C. minuta* is a Gram-negative, strict anaerobic species that belongs to the family of *Christensenellaceae* and the phylum of *Firmicutes* [25]. It is a non-spore-forming, non-motile bacterium of short rods. Most publications state it is Gram-negative; however, Gram-positive dying behavior, despite having a Gram-negative cell wall was observed by Alonso et al. [26]. The main general characteristics of *C. minuta* has been collected in Table 1.

**Table 1.** General characteristics of *Christensenella minuta*.

| Characteristic | Specific to *C. minuta* |
|---|---|
| Morphology | non-spore-forming, non-motile short rods |
| Gram staining | Gram-negative (however, Gram-positive dying behavior was observed as well) |
| Oxygen sensitivity | not extremely oxygen-sensitive |
| Beneficial metabolic relationships | *Methanobacteriaceae* *M. smithii* *Oscillospira* |

*C. minuta* was the first species described in the new family *Christensenellaceae* in 2012 by Morotomi et al. [27]. According to research performed on healthy volunteers in 2014, the bacterium was identified as the most heritable gut microbe in humans, in which its presence is mainly determined by genetic background [28]. *C. minuta* seems to play a major role in the development of a healthy gut microbiome coexisting with other important microbes, and missing in many chronically ill patients. Many observational studies confirmed that the relative abundance of *C. minuta* is associated with a lean body type with a low host body mass index (BMI), and is also influenced by age, diet, and genetics [29–33]. Brooks et al. [34] reported a higher relative abundance of *Christensenellaceae* in women compared to men. Human longevity also influences the relative abundance of *Christensenellaceae*, as it is greater in centenarians compared to younger individuals [35,36]. Based on previous observations, Goodrich et al. [22] successfully proved the weight gain reducing effect of *C. minuta*. They monitored germ-free mice colonized with human fecal microbiota collected from obese donors with no detectable sign of the genus. After a 21-day treatment, the mice receiving the *C. minuta* treatment weighed significantly less than those receiving the unmodified stool. Strong associations were observed between *Christensenellaceae* and BMI as well, and the strain was significantly enriched in individuals with low-normal BMI compared to overweight BMI. They also found that the impact of host genetics on the abundance of *Christensenellaceae* is independent of BMI [28].

According to recent research by Mazier et al. [37], *C. minuta* limits body weight gain and regulates several metabolic markers such as glycemia and plasma leptin in the diet-induced obesity (DIO) mouse model. Studying the mechanism of action, they assume it plays a keystone role in gut microbiome restoration in obese individuals, maintains proper intestinal epithelial integrity, reduces hepatic triglycerides and free fatty acid accumulation, and limits the risk of obesity. Beaumont et al. found a negative relationship between *Christensenellaceae* and abdominal adiposity with the help of dual-energy X-ray absorptiometry in twins from the UK, supporting previous findings from Goodrich et al. [28]. In this context, we can suggest that individuals with *Christensenellaceae* have less cardiovascular risk than those without [38].

A similar observation was found by Hibberd et al. [39], who reported significant negative correlations of *Christensenellaceae* with trunk fat. Moreover, *C. minuta* level is inversely proportional to serum triglycerides according to several studies. Waters et al. summarized that several research groups reported a negative correlation of *C. minuta* abundance with serum lipids such as low-density lipoprotein (LDL) and its component, apolipoprotein B, and total cholesterol levels as well [40].

*Christensenellaceae* have been identified as missing microbes in different inflammatory conditions, suggesting their protective role in the regulation of inflammation. Kropp et al. [41] were the first who showed that *C. minuta* has strong immunomodulatory properties in vitro and in vivo. They analyzed whether *C. minuta* could modulate TNF-α induced secretion of IL-8, a major proinflammatory cytokine in human colorectal adenocarcinoma HT-29 cell line with epithelial morphology and found that IL-8 production decreased by around 50% via the inhibition of the NF-κB signaling pathway. Additionally, *C. minuta* successfully maintained the intestinal barrier of Caco-2 cells after TNF-α exposure,

apparently via the anti-inflammatory action. Finally, they proved in vivo in two different preclinical animal models of acute colitis that *C. minuta* DSM 22607 could reduce colonic inflammation, thus preventing intestinal damage. Their results indicate *Christensenella* as a promising candidate for microbiome-based inflammatory bowel disease (IBD) therapies including Crohn's disease and ulcerative colitis. In contrast with the above-mentioned statements suggesting that increased *C. minuta* levels tend to be beneficial, Pellegrini et al. reported gut dysbiosis with an increased abundance of *C. minuta* in Parkinson's disease patients. This claim would require further investigation [42]. Relation of *C. minuta* levels with different diseases and health conditions are presented in Figure 1.

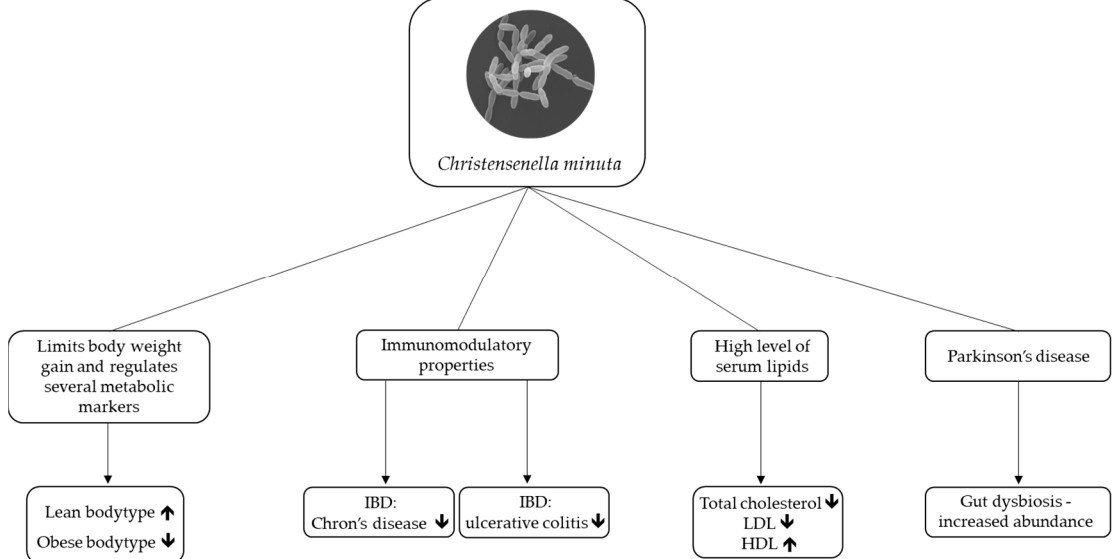

**Figure 1.** Relation of *Christensenella minuta* levels with different diseases and health conditions.

## 3. Aspects of Metabolism

*C. minuta* utilizes various sugars, such as glucose, D-xylose, D-mannose, salicin, L-ramnose, and L-arabinose, and produces volatile fatty acids (VFAs) as fermentation end products. However, it is not able to utilize maltose, lactose, trehalose, sucrose, D-sorbitol, raffinose, D-mannitol, melesitol, or cellobiose. *C. minuta* has positive enzymatic activity in the cases of β-galactosidase, naphthol-AS-BI-phosphohydrolase, α-arabinosidase, β-glucosidase, and glutamic acid decarboxylase [25]. The main end products of glucose metabolism are acetic acid and butyric acid [27]. It has positive enzymatic activity for β-galactosidase, naphthol-AS-BI phosphohydrolase, α-arabinosidase, β-glucosidase and glutamic acid decarboxylase. *Christensenella minuta* is negative for aesculin hydrolysis, catalase, gelatin hydrolysis, indole production, nitrate reduction, oxidase and urease. Acid is produced from glucose, salicin, D-xylose, L-arabinose and L-rhamnose. Acid is only produced weakly from D-mannose, and not produced from cellobiose, D-mannitol, melezitose, raffinose, D-sorbitol, trehalose, glycerol, lactose, maltose, or sucrose at all. It also produces key enzymes, which are responsible for the degradation of chitin derivatives. Borrelli et al. observed that the relative abundance of *C. minuta* was significantly increased in laying hens fed with *Hermetia illucens* larvae containing a high amount of chitin. Chickens have chitinase enzymes in their digestive system, thus they can break down chitin-containing insects in dimers of GlcNAc to produce chitooligosaccarides. The chitin content in larvae may serve as a substrate for the gut microbiome, influencing the composition and the microbial fermentation metabolites [43]. They proved that *C. minuta* can use the produced chitooligosaccarides for further short-chain fatty acid production [43]. Glucosamine and chondroitin are two frequently used supplements. They are poorly absorbed and therefore metabolized in the gut microbiota. Navarro et al. investigated how glucosamine and chondroitin (G&C) treatment changes the gut microbiota. As the result of a 14-day treat-

ment with G&C, the level of the *Christensenellaceae* family and *Bifidobacterium* significantly decreased [44]. The results of this experiment suggest that the above-mentioned G&C cannot be an adequate source of nutrients for *C. minuta*.

According to some observations *C. minuta* can establish and maintain beneficial metabolic relationships with other bacterial species. It was revealed in the past that the *Christensenellaceae* and *Methanobacteriaceae* bacterial family cooccur in many individuals with lean body types. Ruaud et al. [45] cocultured three *C. minuta* spp. together with *Methanobrevibacter smithii*, and it was observed that *C. minuta* spp. were able to efficiently maintain the metabolism of *M. smithii* via $H_2$ production better than *Bacteroides thetaiotaomicron* did. In a culture with *C. minuta*, $H_2$ utilization by *M. smithii* shifted the metabolic production of *C. minuta's* fermentation toward acetate rather than butyrate. Konikoff et al. [46] reported a possible mutualistic relationship between *C. minuta* and the members of the *Oscillospira* genus, as these bacteria are enriched in lean people. In germ-free mice that were treated with obese donor microbiota spiked with *C. minuta*, *Oscillospira* was found to be enriched compared to those animals that were transplanted only with non-spiked microbiota.

Since its discovery in 2012, *C. minuta* has been considered a strictly anaerobic species. However, Kropp et al. [41] recently characterized the oxygen sensitivity of *C. minuta* and found that the bacterium was not extremely oxygen sensitive, as it was able to tolerate oxygen for at least 24 h. This oxygen tolerance may enable its use in industrial manufacturing processes, a possible practical consideration in the large-scale cultivation of *C. minuta*.

## 4. Possibilities of Large-Scale Cultivation

Over the last decades, scientists have studied how humans can isolate new microbiological organisms that are not available in nature for further processing, as a significant amount of the bacteria in our environment remain uncultivated. In addition, many bacterial strains have no cultivable members and can only be identified by molecular methods to detect their DNA. Several strategies were developed and successfully implemented; for example, targeted bacteria detection in mixed-plate cultures via colony hybridization, culturing in simulated natural environments either with 'helper' strains, or the development of modified culture media preparation techniques [47]. However, most of the uncultivated bacteria still require special strategies and methods for cultivation, as they are unable to grow under conventional conditions. Numerous innovative methods that are used for culturing uncultivated bacteria are obtained from environmental microbiology, as a significant number of uncultivated bacteria can be found in nature. In order to develop a commercially available product from naturally occurring, useful but not yet isolated, bacteria, an own strain isolation is needed. Reusing wastewater for this purpose could be a promising alternative to isolation from human and animal sources, as ethical concerns can be eliminated [48].

Besides its potential, *C. minuta* has not been successfully accumulated in large quantities yet. Although *C. minuta* does not require special maintenance and environmental conditions, it can grow between 25 to 43 °C. The most optimal temperature for growth is between 37 to 40 °C. As for pH, the most optimal pH value for growth is 7.5. However, it is able to grow between pH 6 and 9 [25]. Essential environmental conditions and metabolic properties of *C. minuta* are summarized in Table 2.

The above-mentioned undemanding nature of *C. minuta* can be a possible explanation for the observation of Gao et al. [49]. Their research group installed a microfiltration membrane for sludge anaerobic fermentation and monitored the changes of the VFA composition and the microbial community structure. They observed that the membrane rejected more polysaccharides than proteins. The *C. minuta* levels were significantly enriched, from 6.89% in stage 1 to 14.06% in stage 2, which can be explained by the different retention rates of polysaccharides and proteins in the reactor. They also found that another polysaccharide-using bacteria genus, *Parabacteroides*, did not benefit from any advantage from the membrane separation. The relative abundance of *Parabacteroides* reduced by almost 50%, from 8.85% in stage 1 to 4.53% in stage 2. Thus, we can suggest that the type of

polysaccharides retained in the reactor are crucial, as they promote or limit the growth of bacteria. This finding has great potential for the industrial scale-up of *C. minuta*, as it could be a simple and cost-effective solution for increasing the yield of the bacterium. Following this line of thought, utilizing the sludge of different food industrial units; for example, canning factories could be a promising possibility to develop a reliable method for the cultivation of *C. minuta*. Reusing the wastewater contributes to a more sustainable economy. As a major challenge, the whole process must be performed under GMP conditions that are regulated and inspected at the national level. Against pure cultivation, the main concern about large-scale cultivation from sludge is the purification process, which is a strict requirement in the food and pharmaceutical industry as well [19].

**Table 2.** Essential environmental conditions and metabolic properties of *Christensenella minuta*.

| Conditions | Specific to *C. minuta* |
|---|---|
| Optimal pH | 7.5 |
| Optimal temperature | 37–40 °C |
| Oxygen sensitivity | Anaerobic (it is able to tolerate oxygen for at least 24 h) |
| Utilized sugars | glucose, D-xylose, D-mannose, salicin, L-ramnose, and L-arabinose |
| Sugars that cannot be utilized | maltose, lactose, trehalose, sucrose, D-sorbitol, raffinose, D-mannitol, melesitol cellobiose |
| Enzymatic activity | β-galactosidase, naphthol-AS-BI-phosphohydrolase, α-arabinosidase, β-glucosidase, and glutamic acid decarboxylase |

## 5. Strategies for Improved Oral Delivery

The development of probiotic preparations in the last few decades mainly specialized in *Lactobacillus* and *Bifidobacterium* species. Probiotic supplements are mainly sold as powders, tablets, or capsules, which offer easy administration, long shelf life, and high patient compliance. Newly discovered next-generation probiotics have great potential in the prevention and therapy of various conditions. After cultivation, the bacterial cell biomass must be freeze-dried under strictly anaerobic conditions and different microbial quality control tests must be performed, such as microbial purity and viable cell counting. As there is no product containing *C. minuta* on the market so far, we can only assume the most optimal innovative pharmaceutical dosage form based on the results published yet. Being an aerobic microorganism, numerous factors, including oxygen exposure, osmotic pressure, desiccation, extreme changes in temperature, and humidity, can affect the viability of the cells during processing, as well as storage. Moreover, the tough gastrointestinal circumstances, characterized by low pH in the stomach and bile salts in the small intestine are harmful to many microbes. Next-generation probiotics are usually oxygen-sensitive and they do not tolerate the well-known product processing methods; protection is necessary during delivery and stomach transit. Therefore, it is necessary for new processing methods to be developed in order to formulate effective preparations from NGP [24,50].

According to recent investigations, microencapsulation is one of the most promising and efficient techniques for the enhancement of viable probiotic bacteria. The encapsulation matrix can reduce cell injury or death via protecting them from heat during formulation. Bacterial stability and viability within the product can be further increased if the matrix contains various reducing agents such as sulphur compounds, glutathione, sodium thioglycolate, cysteine, dithiothreitol, or sodium dithionite [51]. Encapsulation also ensures sufficient integrity for the cells in the gastrointestinal tract to reach their target area without any degradation [52]. One of the most popular encapsulation polymers is the natural polysaccharide sodium-alginate due to its useful benefits such as biocompatibility, biodegradability, and biosafety. In the presence of divalent cations such as calcium, alginate forms insoluble ionic cross-linked particles with different diameters. Moreover, these complexes can protect target drugs as they are insoluble in the harsh acidic environment

of the stomach [53]. Many research groups have successfully encapsulated the two most common probiotic strains, *Lactobacillus* and *Bifidobacterium*, using calcium ions, thus considering that the encapsulation of *C. minuta* could be a potential way to introduce it to the market [54,55]. Chitosan is a well-known mucoadhesive linear polysaccharide which consists of glucosamine units. Similar to alginate, in the presence of anions and polyanions, the units can be polymerized by the formation of cross-links. A reliable method of the administration of viable bacterial cells to the gastrointestinal tract is the encapsulation of probiotic cells by coating them with a mixture of alginate and chitosan. This formulation technique provides cell protection in GI conditions [56]. On the other hand, the use of chitosan has some disadvantages. Mortazavian et al. [57,58] found that encapsulation with chitosan did not enhance the viability of bacterial cells due to its antimicrobial effect. This suggests that chitosan should be used more as a coating agent, rather than as an encapsulation polymer. Another natural polymer, which can be used in the food and pharmaceutical industry for encapsulation, is k-Carrageenan. The gelation occurs at room temperature when the cells and the polymer are mixed together [59]. The addition of potassium ions can stabilize the formed microparticles. According to the evaluation of Dinakar et al., the probiotic bacterial cells remain in a viable environment when encapsulated with k-carrageenan [60]. The microbial polysaccharide gellan gum can be a potential polymer as well. It was first derived from *Pseudomonas elodea*, which consists of a repeating unit of four monomers, namely glucose, glucuronic acid, glucose, and rhamnose [61]. The encapsulation of probiotic cells with gellan gum can be carried out by using a mixture of xanthan–gelan gum. As compared to the use of alginate for encapsulation, high resistance toward acid conditions is exhibited using a combination of xanthan–gelan gum.

The Spray-drying method is also becoming more and more popular recently. In this method, a solution consisting of the living probiotic cells and the dissolved polymer matrix is prepared first. One of the most commonly used polymer matrices are gum arabic and starch, as during the spray-drying process, they are able to form spherical- shaped microparticles [61]. An additional coating layer can be applied to protect the particles from the acidic environment of the stomach and to reduce the adverse effects of bile salts as well. Spray drying has both advantages and disadvantages. One of the main advantages of the technique is its rapidity and cost-effectiveness. This method can be suitable for industrial applications as well. One of the major disadvantages of spray drying is the relatively high temperature used during the procedure, which is not optimal for cell viability. This difficulty can be solved via a different formulation process called freeze drying or lyophilization. During freeze drying, the water content of the formulation is first removed under a vacuum via sublimation. It is followed by desorption after the probiotic cells and the carrier material are frozen as well at −20 to −30 °C. Cryoprotectants can be added after lyophilization to maintain and stabilize the probiotic viability during storage. The most commonly used cryoprotectants are skim milk, lactose, trehalose, sucrose, and sorbitol. Lyophilized products offer longer shelf life compared to spray-dried ones [62].

Two promising but less widespread novel delivery systems for the administration of probiotic species are microdevices and polymeric fibers. Microdevices are mostly designed for therapeutic drug delivery in the gastrointestinal tract, but the devices may also be suitable for carrying probiotics. They usually include micropatches, microwells, or microcontainers in the size range of 100–300 µm, typically in a square or spherical shape [63]. Non-biodegradable and biodegradable materials, e.g., polylactic acid can also be applied as vehicles of microdevices; however, their use for this purpose is under testing. The microdevice-loading techniques are novel and not well-tested for the loading of microorganisms. After the loading of the desired content into a microdevice, a coating step usually follows to create a lid and seal off the device [24].

Polymeric fibers have mostly been studied as carriers for oral drug delivery, although they could be suitable for carrying probiotics as well. Fibers are generally produced by electrospinning, a procedure that utilizes an electric field to alter and produce long, thin polymeric structures from a solution pressed out of a syringe. Desiccated or active

microbes can be added to the solution, which usually consists of polyvinyl alcohol, alone or combined with other materials, such as alginate, hence creating microbe-embedded fibers with increased viability protection of the microbes in the stomach [24].

The above-mentioned new and innovative methods, according to the literature, could all offer protection for the sensitive microbes; thus, they are potential carriers of *C. minuta*.

## 6. Conclusions

A great amount of research and investigation into the benefits of the gut microbiome in both human and animal health leads to the development of next-generation probiotics from newly isolated microorganisms. These improvements present major challenges for scientists, and industries as well. *C. minuta* is one of these potential next-generation probiotics with many advantages. Despite not requiring any special maintenance and environmental conditions, still, *C. minuta* has not been cultivated effectively in larger quantities. According to Gao et al. [64]'s research in 2019, a promising key for the production could be sludge fermentation in an anaerobic membrane bioreactor. The oral administration of *C. minuta* with different prebiotics; for example, fructans, galacto-oligosaccharides, or even chitooligosaccharides could be a possible symbiotic, as they may serve as a substrate for the gut microbiome. Similar to most probiotics on the market, *C. minuta* could be used in combination with other bacteria to enhance its positive effects, as it is able to develop beneficial co-cultures with several strains. *C. minuta* has been associated with lean body types and lower body weight. In a test conducted on 977 volunteers, humans with elevated levels of *C. minuta* in their gut tend to have lower body mass index than those with low levels. According to scientific literature, the presence of *C. minuta* is significant for a wide range of health benefits, and its absence may be associated with several diseases or pathological conditions. Besides the clear correlation between high levels of *C. minuta* and lean body types, gastrointestinal diseases can be observed in the absence of it, suggesting that it may also play a protective role in the regulation of inflammation. However, not only a high quantity of *C. minuta*, but the opposite, a reduced number of the bacterium can be beneficial as well. For example, in animal husbandry, it could be useful in the weight gain of various livestock. Considering these aspects, *C. minuta* has plenty of possibilities, both in the pharmaceutical industry and animal husbandry.

**Author Contributions:** I.B. and I.P. have designed the manuscript. Á.P. and D.K. have written the manuscript. Z.S., P.F., Z.U., G.K. and I.N. have helped revising the scientific literature. All authors have read and agreed to the published version of the manuscript.

**Funding:** The research was supported by the Thematic Excellence Program (TKP2020-NKA-04) of the Ministry for Innovation and Technology in Hungary. Project no. TKP2021-EGA-19 has been implemented with the support provided from the National Research, Development and Innovation Fund of Hungary, financed under the TKP2021-EGA funding scheme. The work/publication is supported by the GINOP-2.3.4-15-2016-00002 project. The project is co-financed by the European Union and the European Regional Development Fund. The work/publication is supported by the GINOP-2.3.1-20-2020-00004 project. The project is co-financed by the European Union and the European Regional Development Fund.

**Institutional Review Board Statement:** Not applicable.

**Informed Consent Statement:** Not applicable.

**Data Availability Statement:** No new data were created or analyzed in this study. Data sharing is not applicable to this article.

**Conflicts of Interest:** The authors declare no conflict of interest.

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
