# Peer review of "Scientific and Pharmaceutical Aspects of Christensenella minuta, a Promising Next-Generation Probiotic"

_fermentation, doi:10.3390/fermentation9080767_

Round 1

Reviewer 1 Report (Previous Reviewer 2)

In the manuscript “Scientific and pharmaceutical aspects about Christensenella minuta, a promising next generation probiotic” (ID 2508747), Ágota Pető, Dóra Kósa and their colleagues summarized the facts and perspectives about Christensenella minuta. They described the relation of Christensenella minuta level with different diseases and health conditions and the essential environmental conditions and metabolic properties of C. minuta. Furthermore, their summarized different processing methods for pharmaceutical aspects. In one word, C. minuta has plenty of possibilities both in the pharmaceutical industry animal husbandry.

The main concerns:

1) In Part 5, strategies for improved oral delivery, they summarized information of many advantages in the probiotics oral delivery systems. What do these methods have to do with C. minuta? That is, why to summarize these methods. This point is not very clear.

2) There are many strange points shown in this manuscript, which may be produced with some constraint condition in the references. These points are easy for readers to misunderstand. For example, line71-72, “At the same time, their widespread administration made them more resistant against certain diseases”.

3) Part 4, possibilities of large-scale cultivation, Gao used sludge anaerobic fermentation system to enrich C. minuta from the mixed bacterial population, which usually used in bacterial isolation process or wastewater treatment, while pure cultivation is needed in food and pharmacology industry. These are two totally different technologies in microbiology field.

4) In the first part, we can not clearly see what mechanism or mode of action C. minuta can achieve the remission or treatment of disease, whether it can be shown by a pattern diagram, rather than just looking at the abundance of C. minuta in disease or healthy people.

Minor points:

1) The format of Table 1 and Table 2 is not standard, and the format of the three-line table should be adopted.

2) Are there any relevant clinical studies focusing on C. minuta?

3) Whether you can use the form of graphs or tables to more intuitively show the metabolic activity of C. minuta.

Author Response

Reviewer 2 Report (New Reviewer)

The Manuscript "Scientific and pharmaceutical aspects about Christensenella minuta, a promising next generation probiotic" was written in a synthetic and coherent manner and is of great interest to the reader. However, there is a lack of literature to support many of the claims, as well as a clear and concise conclusion.  

Please see my specific comments below.

Major comments:

-The introduction needs more citations to support the content

- The bibliography should be reviewed, there are many microorganisms without italics...

-The conclusions should be restated to be concise and clear. All kinds of major information should be included in the main text

Minor comments:

Line 69. The name of the microorganisms in the titles must be complete

Line 100, 103, 105,110, 114 and 122. Use the journal format for references. Alfonso et al. [20]

Line 111. A space must be removed

Line 132 and139.Use the journal format for references.

Line 152. C. minuta instead of Christensenella minuta

Line 173 and 189. As this microorganism has not been previously named, it must be written in full

Line 180, 189, 199 and in all the manuscript. Use the journal format for references.

Line 182. Add a comma after G&C

Line 184. G&C instead of glucosamine and chondroitin. This term has already been used, so from then on it should be used in this way.

Line 189 and 190. C. minuta spp. please revise

Line 227. The name of the microorganisms in the table must be complete

Line 229. Use the journal format for references.

Line 252. A space must be removed

Round 2

Reviewer 1 Report (Previous Reviewer 2)

In the manuscript “Scientific and pharmaceutical aspects about Christensenella minuta, a promising next generation probiotic” (ID 2508747), Ágota Pető, Dóra Kósa and their colleagues summarized the facts and perspectives about Christensenella minuta. They described the relation of Christensenella minuta level with different diseases and health conditions and the essential environmental conditions and metabolic properties of C. minuta. Furthermore, their summarized different processing methods for pharmaceutical aspects. In one word, C. minuta has plenty of possibilities both in the pharmaceutical industry animal husbandry.

Author Response

Dear Reviewer,

First of all, we would like to express our sincere appreciation for the accurate critical review of our manuscript. We appreciate the time and effort that you have dedicated to providing your valuable feedback on our manuscript. We are grateful for the insightful comments on our paper.

Reviewer 2 Report (New Reviewer)

I appreciate the efforts of the authors to improve the quality of the manuscript based on the reviewers' comments, but there are still errors that need to be corrected.

The new bibliography must have the format of the journal

Table 1. C.minute must be written completely

line 230. Add a period after Gao et al.

Species of microorganisms must be written in lower case.

The bibliography must be corrected and reviewed in its entirety. There are citations that are not in the text and other citations from the text are not in the bibliography. In addition, new citations must be added to the bibliography and reorder it

Author Response

Dear Reviewer,

First of all, we would like to express our sincere appreciation for the accurate critical review of our manuscript. We appreciate the time and effort that you have dedicated to providing your valuable feedback on our manuscript. We are grateful for the insightful comments on our paper. We have been able to incorporate changes to reflect all of the suggestions provided. We have listed the changes of the manuscript. Here is a point-by-point response to your comments and concerns. The changes are highlighted green in the manuscript.

Comment 1: The new bibliography must have the format of the journal.

Response 1: Thank you for the observation!  Sorry for the inconvenience, the bibliography has been thoroughly checked again and any remaining errors have been corrected. The new bibliography has been formatted according to the requirements and official format of the journal.

Comment 2: Table 1. C. minute must be written completely

Response 2: Thank you for the observation! It has been corrected to Christensenella minuta.

Comment 3: line 230. Add a period after Gao et al.

Response 3: Thank you for the observation! We added a period after Gao et al.

Comment 4: Species of microorganisms must be written in lower case.

Response 4: Thank you for the observation! We corrected the mistakes.

Comment 5: The bibliography must be corrected and reviewed in its entirety. There are citations that are not in the text and other citations from the text are not in the bibliography. In addition, new citations must be added to the bibliography and reorder it.

Response 5: Thank you for the observation!  Sorry for the inconvenience, the bibliography has been thoroughly checked again and any remaining errors have been corrected. Citations added to the text subsequently were not indeed in the bibliography by mistake, thus the new citations were added to the bibliography and has been reordered as well.

This manuscript is a resubmission of an earlier submission. The following is a list of the peer review reports and author responses from that submission.

Round 1

Reviewer 1 Report

Dear authors

This review covers an interesting and promising next generation probiotic, Christensenella minuta.

The manuscript contains a significant amount of useful and informative research but overall it can be significantly improved.

Title The term "Facts" in the title is unscientific

Abstract

Is poorly written. Starting with the first sentence, it is too long and poorly written creating a very weak opinion for the reader for the remainder of the review. Which itself is riddled with poor English throughout.

The Tables and Figure in the review are lacking in quality and are too simplistic to warrant publication in their current form.

Table 2 does not list the crucial environmental factor, relationship with oxygen.

Section 4 is the key part of the review but it is disappointingly short on detail

Section 5 to me is of less importance but is much longer and could do which more structure.

Overall the review has promise but I recommend significant more work is required to do the topic justice and make the review of publishable standard.

Quality of scientific writing needs to be significantly improved.

Reviewer 2 Report

In the manuscript “Facts and perspectives about Christensenella minuta, a promising next generation probiotic” (ID 2342651), Ágota Pető, Dóra Kósa and their colleagues summarized the facts and perspectives about Christensenella minuta. They described the relation of Christensenella minuta level with different diseases and health conditions and the essential environmental conditions and metabolic properties of C. minuta. Furthermore, their summarized different processing methods for pharmaceutical aspects. In one word, C. minuta has plenty of possibilities both in the pharmaceutical industry animal husbandry.

The main concerns:

1) In introduction part, the authors described the common sense about what are probiotics, prebiotics and synbiotics, and in Part 5, strategies for improved oral delivery, they summarized information of many advantages in the probiotics oral delivery systems. These are unrelated to the main topic of this manuscript, C. minuta.

2) There are many strange points shown in this manuscript, which may be produced with some constraint condition in the references. These points are easy for readers to misunderstand. For example, line75-76, “their widespread consumption made them less effective against certain disease”. Line 79-81, “Administration of commonly used traditional probiotics such as Bifidobacterium spp. and Lactobacillus spp. does not seem effective against the abovementioned specific disorders”.

3) Part 3, possibilities of large-scale cultivation, Gao used sludge anaerobic fermentation system to enrich C. minuta from the mixed bacterial population, which usually used in bacterial isolation process or wastewater treatment, while pure cultivation is needed in food and pharmacology industry. These are two totally different technologies in microbiology field.

4) Figure 1 is too simple to show the connection between the bacteria and the host. Whether you can we add more details to further optimize the picture.

5) The format of Table 1 and Table 2 is not standard, and the format of the three-line table should be adopted.

6) In the second chapter C. minuta as a potential next generation probiotic, many study cases are described. Can you add details of the strains? For example, what is the strain name of the microorganism used?

7) Are there any relevant clinical studies focusing on C. minuta?

Minor points:

1) The format of in vitro and in vivo in line 145 on page 4 is not standard.

2) In the third chapter Aspects of metabolism, a sentence appears three times that is “It has positive enzymatic activity for β-galactosidase, naphthol-AS-BI phosphohydrolase, α-arabinosidase, β-glucosidase and glutamic acid decarboxylase”. It is not very appropriate to write like this.

3) Whether you can use the form of graphs or tables to more intuitively show the metabolic activity of C. minuta.